# Molecular Ecological Network Complexity Drives Stand Resilience of Soil Bacteria to Mining Disturbances among Typical Damaged Ecosystems in China

**DOI:** 10.3390/microorganisms8030433

**Published:** 2020-03-19

**Authors:** Jing Ma, Yongqiang Lu, Fu Chen, Xiaoxiao Li, Dong Xiao, Hui Wang

**Affiliations:** 1Low Carbon Energy Institute, China University of Mining and Technology, Xuzhou 221008, China; jingma2013@cumt.edu.cn; 2School of Environmental Science and Spatial Informatics, China University of Mining and Technology, Xuzhou 221116, China; TS18160128P31@cumt.edu.cn (Y.L.); lixiaoxiao@cumt.edu.cn (X.L.); wanghuei@cumt.edu.cn (H.W.); 3State Key Laboratory of Coal Resources and Safe Mining, China University of Mining and Technology, Xuzhou 221116, China; xd@cumt.edu.cn

**Keywords:** disturbed mining areas, soil microbial community, microbial network interactions, network topology, keystone taxa, soil resilience

## Abstract

Understanding the interactions of soil microbial species and how they responded to disturbances are essential to ecological restoration and resilience in the semihumid and semiarid damaged mining areas. Information on this, however, remains unobvious and deficiently comprehended. In this study, based on the high throughput sequence and molecular ecology network analysis, we have investigated the bacterial distribution in disturbed mining areas across three provinces in China, and constructed molecular ecological networks to reveal the interactions of soil bacterial communities in diverse locations. Bacterial community diversity and composition were classified measurably between semihumid and semiarid damaged mining sites. Additionally, we distinguished key microbial populations across these mining areas, which belonged to *Proteobacteria, Acidobacteria, Actinobacteria*, and *Chloroflexi*. Moreover, the network modules were significantly associated with some environmental factors (e.g., annual average temperature, electrical conductivity value, and available phosphorus value). The study showed that network interactions were completely different across the different mining areas. The keystone species in different mining areas suggested that selected microbial communities, through natural successional processes, were able to resist the corresponding environment. Moreover, the results of trait-based module significances showed that several environmental factors were significantly correlated with some keystone species, such as OTU_8126 (*Acidobacteria*), OTU_8175 (*Burkholderiales*), and OTU_129 (*Chloroflexi*). Our study also implied that the complex network of microbial interaction might drive the stand resilience of soil bacteria in the semihumid and semiarid disturbed mining areas.

## 1. Introduction

Coal mining activities have resulted in surface subsidence, and have made the ecological environment more fragile by creating huge, overburdening dumps and voids [1]. Recently, increasing attention has been paid to the influences of coal-mining subsidence on the ecological environment [2]. The soil problems caused by coal mining have become increasingly prominent and already have been an important research topic in mining environmental ecology. Mining activities severely disrupt land soils, resulting in the deterioration of the existing local ecosystems, such as destroying or degenerating essential properties in the original soils [3]. The physical and chemical properties of existing soil and microbial community characteristics have been seriously disturbed, and the quality of reclaimed or restored soil has been quite poor [4]. Due to the protection of cultivated land and food security, the ecological restoration of mining areas with high groundwater levels has focused on soil reclamation in Eastern China. In the northern-western part of China, rapid and effective ecological restoration is also in critical demand in order to ensure the sufficient management of semiarid, damaged mines.

Using soil microbes is important in order to stimulate an ecosystem’s resilience. Assessment of the diversity and activity of the soil microbial community is essential to evaluate the success of reclamation or restoration. However, few studies have been conducted on the soil microbial community diversity where there is a high groundwater level, or in semiarid damaged mining areas [5,6,7]. In this study, we identified the dominant bacteria, which was critical to enhance our understanding, and determined the ecological attributes of soil bacterial communities, which are abundant and ubiquitous in the soil at different mining areas. Understanding the ecological attributes of dominant bacteria will increase our capacity to successfully cultivate them, which is critical to successful restoration and reclamation progress in mining areas [8,9]. In addition, understanding how soil bacterial communities vary across space and how they respond to mining activities is also important for restoration ecology [10]. For example, by locating and identifying some dominant taxa, which tend to prefer special environmental conditions, such as mine cracks and surface subsidence, we can forecast their distribution and enrich them to enhance the ecological restoration capacity of damaged mines. Thus, a better understanding of dominant soil microbial taxa in the mining areas would improve our ability to manage soil bacterial communities and promote their functional abilities.

Microbial biodiversity includes the number of species, their abundance, and the complex interactions among different species [11]. In the environmental habitats, massive microbial species interact with each other to form complex ecological networks [12]. It is important to understand microbial structural and functional effects, and the changes in microbial biodiversity, which might be elucidated through the networks of interacting species. Additionally, it is essential, in studying microbial biodiversity, to elaborate on and analyze the interactive network structures, as well as to understand the underlying mechanisms. Therefore, in microbial ecology, the ecological networks of biological communities have gained attention. However, it is still difficult to determine the network structures and their relationships with environmental changes in microbial communities [13]. The microbial community assembly process significantly affects the structure of microbial community, and the selection process acts as one of the ecological processes controlling the assembly of the microbial community. Moreover, this microbial interaction, which can be seen as a kind of selection, provides some contributions to the microbial community assembly process [14]. Therefore, researchers have increasingly studied microbial networks in diverse environments [15]. These microbial interactions have been emphasized as being crucial to our understanding of the dynamics of microbial community assembly alongside climate change [16]. Although some studies have investigated the changing microbial interactions in response to different environmental disturbances, few studies have revealed how microbial interactions vary in subsidence areas or damaged mines, especially in different locations. Furthermore, deficiencies exist in how coal-mining activities have changed the structure of soil bacterial communities and their interactions. Fortunately, in recent years, numerous studies have verified the effect of land reclamation on soil bacterial communities after coal-mining disturbances [17]. Moreover, some studies have focused on the relationship between changes in soil bacterial communities and surrounding environmental factors. They discovered that changes in soil bacterial communities were closely associated with soil properties, enzyme activities, and various types of vegetation cover. Some studies also reported on how the structure of soil bacterial communities and their diversity changed after coal-mining disturbances [7,17,18].

The interactions of different microbial populations in a community play critical roles in determining the functioning of an ecosystem, but little is known about the network interactions in the microbial community, primarily because of the lack of appropriate experimental data and computational analytic tools [19]. In recent years, high-throughput metagenomics technologies have rapidly produced a massive amount of data, but one of the greatest difficulties in managing these data is deciding how to extract, analyze, synthesize, and transform such a vast amount of information into biological knowledge [20]. This study provided a novel conceptual framework to identify microbial interactions and key populations based on high-throughput metagenomics sequencing data. The availability of massive, community-wide, and replicated meta-genomic data from different mining areas has provided an unprecedented opportunity to analyze network interactions in a microbial community [21].

By combining massive data, we introduced molecular ecological network (MEN) construction methods and the statistical analysis of bioinformatics to explore the controlling factors affecting the distribution of microbial communities with high groundwater levels and in semiarid mining areas. Furthermore, we explained the relationships between soil microbial communities and explored some microbial keystone groups that could respond to and adapt to environmental changes. On the other side, the network approaches might provide a new way to improve the ecological diversity and ecosystem services in subsided or reclaimed mining areas, through a better decision-making, based on a more complete evaluation. The appearance of molecular biological techniques provides new methodologies to construct large-scale replicated networks, although system-level responses to change remain mostly unexplored. We addressed three hypotheses in the current study: First, network properties differed significantly among mining habitats on a large scale geographic scale level. Second, the soil properties that correlated with keystone bacterial communities were different across the mining areas. Third, the microbial distribution patterns across spatial distance, and the interactions of bacterial communities among mining areas might drive different soil resilience levels in future mine restoration and reclamation. Finally, we hope this study will aid in defining the recovery resilience of a damaged mine ecosystem from the perspective of a microbial MEN, and revealed the development pattern of microbiome, and the ecological restoration elastic enhancement mechanism.

## 2. Materials and Methods 

### 2.1. Study Sites, Soil Sampling, and Measurment

The Peibei (PB) coal-mining area (34°13′39”N–34°26′16”N, 117°06′21”E–117°12′16”E) is located in Northern Anhui and Jiangsu Province (Figure 1). The study area has a warm temperate zone with a semihumid monsoon climate and four distinctive seasons. The area has an annual average temperature (AAT) of 14 °C and an annual average precipitation (AAP) of 800–930 mm, which is a representative semihumid area in East China. The soil type was haplic brown, and the sampling sites were in the subsided mining areas. The Zoucheng (ZC) coal-mining area is located in Shandong Province (35°8′12”N–35°32′54”N, 116°46′30”E–117°28′54”E), which is situated in a warm temperate monsoon climate zone (Figure 1). This Geographically representative semihumid area of Eastern China has an annual rainfall of 777.1 mm, and an annual average temperature of 14.1 °C. The soil type is fluvo-aquic soil. The samples were collected from the reclaimed farmland in the mining area. The Yangquan (YQ) coal-mining area (113°15′E–113°18′ E, 38°01′N–38°03′N) is located in Shanxi Province, China (Figure 1). It has a continental climate, with an annual average temperature of 8.7 °C, and an annual rainfall between 450 and 550 mm, which is classified as a representative semiarid area of typical geographical environment in Western China. The region is at the southern end of the Loess Plateau, and the main soil type is calcareous cinnamon soil. Moreover, the ecological environment has been seriously damaged, with frequent land cracks and an exposed vegetation root system along the surface. We collected the soil samples from the damaged areas. The Datong (DT) coal-mining area (39°53′24″N–40°10′00″N, 112°52′13″E–113°32′35″E) is also located in Shanxi Province, China (Figure 1). This area used as the representative semiarid area in Northern-Western China, has a continental climate and a mean annual temperature of 6.4 °C. The mean annual precipitation is 384.6 mm, with precipitation mainly occurring from June to September. The collected soil type is loess and from the damaged mining areas. Using the diamond sampling method, each soil sample was composed by 4 soil samples collected from plots those were 9 m^2^ in size in the four mining areas.

From June to August 2018, we collected approximately 500 g of surface (0–10 cm) soil from 14 discrete locations in the each mining area (Figure 1). We stored about 20 g of soil at −20 °C for subsequent analysis of the microbial diversity. The remaining soil was air-dried and homogenized to pass through a 2 mm sieve. We measured the soil pH and electrical conductivity (EC) values using a pH meter and a conductivity meter, respectively (PHC-3C, DDS-307A, Shanghai leici, China). We measured the soil organic matter (SOM) according to colorimetrical methods using hydration heat during the oxidation of potassium dichromate. We also analyzed soil ammonium nitrogen (AN) using the potassium chloride-ultraviolet spectrophotometry method, and measured the nitrate-nitrogen (NN) content by calcium chloride-ultraviolet spectrophotometry. We measured the available phosphorus (AP) using the hydrochloric acid ammonium chloride method. We analyzed the available soil potassium (AK) by the ammonium acetate–flame photometric method.

### 2.2. DNA Extraction, PCR Amplification, and Illumina MiSeq Sequencing

According to the manufacturer’s instructions, we extracted DNA from 56 soil samples taken from 0.5 g of fresh soil samples using the FastDNATM SPIN Kit for Soil (MP Biomedicals, Solon, OH, USA). We amplified the V4–V5 region of the bacterial 16S rRNA genes using the primer sets 515F (5′-GTGCCAGCMGCCGCGGTAA-3′) and 907R (CCGTCAATTCMTTTRAGTTT). The DNA Gel Extraction Kit (Axygen Biosciences, Union City, CA, USA) was used to pool and purify the polymerase chain reaction (PCR) products. We quantified the purified PCR products using the Quant-iT PicoGreen dsDNA Assay Kit (Invitrogen, Carlsbad, CA, USA). The purified amplicons were paired-end sequenced (2 × 300) on the Illumina MiSeq platform using the MiSeq Reagent Kit V3 (Personalbio, Shanghai, China). We distinguished the sample sequencing data according to the barcode sequence and checked the sequence of each sample for quality control. Then we removed the nonspecific amplification sequences and chimeric with USEARCH (v5.2.236, http://www.drive5.com/usearch/) in QIIME (v1.8.0, http://qiime.org/). We clustered the operational taxonomic units (OTUs) with a 97% similarity cutoff using the UCLUST method in QIIME and used the Greengenes database (release 13.8, http://greengenes.secondgenome.com/) to classify the species [22,23]. We conducted alpha diversity indices to reveal the richness, diversity, and evenness of the OTUs and performed beta diversity analysis online using the open-source platform Metagenomics for Environmental Microbiology (DengLab; http://mem.rcees.ac.cn:8080/). According to the taxonomic results, we constructed an abundance diagram and obtained rich infrared images with Origin 9.1 (OriginLab, Northampton, MA, USA) and R software (https://www.r-project.org/). The principal component analysis (PCA), non-metric multidimensional scaling (NMDS), response ratio calculation (RRC), canonical correspondence analysis (CCA), variation partition analysis (VPA), correlation test, mantel test, and LEfSe (linear discriminant analysis effect size) were also conducted on this platform.

### 2.3. Network Construction and Analysis

On the basis of 16S rDNA sequencing data, we used all the data from the 56 collected soil samples to construct the interaction networks, which we defined as phylogenetic MENs [24]. For these 56 samples, each mining area had 14 samples to establish their own networks.

According to Deng et al. [25], we followed four steps in the construction process: data collection, data transformation, pairwise similarity matrix calculation, and adjacent matrix determination. During the construction, we only used the OTUs (97% sequence identity) occurring in 100% of the total samples for the network computation. Then, we filled the blanks of 0.01 with paired valid values. As recommended, we used Spearman’s Rho to measure the correlation and calculated a similarity matrix. Thereafter, we increased the similarity threshold from 0.01 to 0.99 with intervals of 0.01, and selected an optimal similarity threshold. We determined significant non-random patterns by evaluating whether the spacing of the eigenvalue distribution followed a Poisson distribution. In order to allow for a comparison, we used an identical cutoff of 0.86 to construct the interaction networks for each mining area. We performed network construction and statistical analysis using the existing pipeline available at http://ieg4.rccc.ou.edu/mena. We visualized these networks with Cytoscape 3.7.0 software [26].

### 2.4. Characterization of the Molecular Ecological Networks and Statistical Analysis

We calculated network global properties, including total nodes and links, R2 of power-law, average degree (avgK), and average path distance (GD). Then, we calculated network indices for individual nodes on the pipeline, such as degree and stress centrality. Greedy modularity optimization was presented as a separation method for module separation. In the network, module was defined as a group of OTUs with a high connection among themselves, but few connections were made with OTUs outside the group. Furthermore, modularity (M) was extremely important for system stability [27]. Then we calculated two important parameters, Zi (within-module connectivity) and Pi (among-module connectivity), for the modularity of all the nodes. According to the values of Zi and Pi, we classified the roles of nodes into four categories: peripherals (Zi ≤ 2.5, Pi ≤ 0.62), connectors (Zi ≤ 2.5, Pi > 0.62), module hubs (Zi > 2.5, Pi ≤ 0.62), and network hubs (Zi > 2.5, Pi > 0.62) [22]. Additionally, we fitted three power-law models for the first step of the network statistics. Then, to evaluate the constructed networks, we rewired the network connections and calculated the network properties randomly with 100 permutations between random and empirical networks.

We also calculated the relationships between gene significances (GS) and environmental traits and used the Mantel test to check for correlations between GS and network connectivity. The GS was calculated and defined as the square of the Pearson correlation coefficient (R2) of the OTU abundance profile with environmental traits. We used these correlations between GS and network indices to reveal the internal associations between network topology and environmental traits. During the process, we used the Euclidean distance method. Then we ran the process of module-eigengene analyses on the pipeline. The eigengene analysis was useful in revealing higher-order organizations and to identify key populations based on network topology. In the analysis, we summarized every module through a single value decomposition analysis, which we referred to as the module eigengene. The relative abundance profile of the OTUs within a module could be shown in eigengene. Moreover, the relationships among eigengenes have been visualized as a clustering dendrogram through average-linkage hierarchical analysis [24]. Additionally, we calculated the relationships between traits and modules, which are shown in a heatmap.

## 3. Results

### 3.1. The Taxonomic Composition of Microbial Consortia in Different Mining Areas

We analyzed the taxonomy alpha diversity of the soil microbial communities for 56 soil samples (Table 1). After comparing the Chao and Shannon index for each mining area, we found that soil bacterial diversity in the Zoucheng (ZC) network was the highest and that in the PB network had the smallest value. The results implied that the ZC area had the highest species richness and diversity, whereas Peibei (PB) had the lowest. We used Pielou evenness to measure the heterogeneity of the community. The data in Table 1 showed that the value of ZC was the highest, which indicated that the evenness of its microbial community species was the best. Moreover, the values of the Chao and Shannon index were the smallest in PB, although the value of Pielou evenness was not the smallest.

Overall, the bacterial categories were relatively abundant in the 56 soil samples (Figure 2). *Acidobacteria, Actinobacteria, Bacteroidetes, Chloroflexi, Gemmatimonadetes, Nitrospirae, Planctomycetes*, and *Proteobacteria* accounted for almost 90% of the total sequences in each soil sample. *Cyanobacteria, Candidatus Saccharibacteria* (TM7), *Firmicutes, Thaumarchaeota*, and *Verrucomicrobia* were present in some soil samples with little occupation. The most abundant phylum in the PB, ZC, and Yangquan (YQ) mining areas was *Proteobacteria*, which accounted for 37.42% ± 6.26%, 32.67% ± 4.13%, and 24.80% ± 3.95%, whereas it was *Actinobacteria* (29.68% ± 12.37%) was the most abundant in the DT mining area (Figure 2). *Acidobacteria* was the second most abundant phylum in PB (the proportion was just 13.38% ± 5.63%). However, *Actinobacteria* was found to be the second most abundant phylum in the ZC and YQ networks. Figure 2a,d also showed that the phylum *Cyanobacteria* and *Verrucomicrobia* accounted for more than 1% in the PB mine, whereas TM7 (1.47% ± 2.39%) was the most present in the Datong (DT) mine. Phylum *Thaumarchaeota* appeared in the ZC and YQ mining areas with a proportion of more than 1%. Furthermore, the proportion of *Acidobacteria* trended in the order of ZC > DT >YQ > PB, and *Chloroflexi* had a similar occupation of around 10%. The proportion of *Nitrospirae* was shown to be around 1% in the four mining areas. The proportion of *Bacteroidetes* trended in the order of PB > ZC > YQ > DT in the four areas. Moreover, the proportion of *Gemmatimonadetes* trended in the order of DT > PB > YQ > ZC, and *Planctomycetes* trended in the order of YQ > ZC > PB > DT in the four mining areas. These results suggested that the microbial distribution patterns across spatial distance varied among the four mining areas, which supported half of the third hypothesis.

Based on the community structure, principal component analysis (PCA), non-metric multidimensional scaling (NMDS), and response ratio calculation (RRC) were performed for the bacterial structure comparison of the four mining areas. The results indicated that the differences among the microbial structures and compositions were representative. The colorful dots shown in Figure 3 stand for different samples (or communities). If two dots were closer, it meant that there was a higher similarity between the microbial community structures of the two samples. The results of PCA and NMDS analysis showed that the soil microbial communities from the four mining areas were different, whereas the bacterial composition and structure within each group (mining area) were grouped closely. It can be seen that the dot arrangement of PCA represents the distinctive pattern along the vertical axis, according to the location order of PB, ZC, YQ, and DT. However, the NMDS analysis showed that the groups followed a different pattern, with the horizontal axis separating the four groups along the spatial order of PB, ZC, YQ, and DT.

### 3.2. Topological Properties of MENs in Different Mining Areas

In recent years, the method of network analysis has been proposed as a new way to explore interaction patterns of complicated data sets, which may provide more information than alpha–beta diversity analysis. Therefore, in this study, to derive a better understanding of big differences between the composition and abundance of soil bacterial communities in mining areas, we used network analyses to explore the associations between soil bacterial taxa in the mining sites.

In MENs, the microbial species (meaning nodes) are linked by pairwise interactions (meaning links), which may reveal some of the biological interactions in the ecosystem. In this study, we individually constructed four networks from different mining areas. We investigated some important general network topological features, such as the scale free, small world, or modular, to understand the differences among these MENs. Table 2 showed that their connectivity followed the power law, and that the network connectivity (or degree) in the four constructed MENs was fitted well with the power-law model (*R*^2^ values of 0.837–0.931, respectively). The results revealed that all the curves of the network connectivity distribution were fitted well with the power-law model, which was indicative of the scale-free networks. Furthermore, the average clustering coefficients and path distances were also different from those of the corresponding random networks (Table 2).

Table 2 showed that the average clustering coefficients (avgCC) of the PB, ZC, YQ, and DT networks were 0.314, 0.258, 0.158, and 0.184, respectively. The average degrees (avgK) of the PB, ZC, YQ, and DT networks were 10.363, 3.894, 1.976, and 2.902. The average path distances (GD) of the PB, ZC, YQ, and DT networks were 3.334, 7.725, 3.975, and 7.802, which were close to the logarithms of the total number of network nodes, suggesting that the four MENs had the typical property of a small world. Deng et al. have reported that a higher avgK is indicative of a more complex network and that a small GD means that nodes in the network are closer [25]. This information shows that the PB network was the most complex network, and it could be identified by the highest avgK and shortest GD (Table 2). For modularity, all modularity values ranged from 0.364 to 0.897, which was higher than the modularity values from their corresponding randomized networks. Therefore, all of the constructed MENs appeared to be modular. In the PB, ZC, YQ, and DT networks, we focused on the modules with more than five nodes. As a result, we detected modules 6, 10, 9, and 13 with more than five nodes. The module sizes varied considerably, ranging from 6 to 73 nodes, and the individual modules showed obvious differences. Most importantly, all of the results confirmed that the network properties differed significantly among different mining habitats, which supported the first hypothesis.

### 3.3. Dominant Microbial Taxa across Different Mining Areas

Microbial network structures were distinctly different among the four networks across the different mining areas, and across the semihumid to semiarid locations in China (Figure 4). Figure 4 showed that there were eight phyla in each network with node degrees > 1, namely, *Acidobacteria, Actinobacteria, Bacteroidetes, Chloroflexi, Gemmatimonadetes, Nitrospirae Planctomycetes*, and *Proteobacteria*. As reported, we considered the nodes with higher degrees to be the central nodes in the network structure [28]. Figure 4 also showed that nodes with high connectivity (degree) varied across the mining areas. In the PB network, the top five nodes (60> node degree >40), which might have been the predominant phylum, belonged to *Acidobacteria* (OTU_24020, OTU_19752 and OTU_20695) and *Gemmatimonadetes* (OTU_30606 and OTU_23181; Appendix A). In the ZC network, *Acidobacteria* (OTU_8126, OTU_34138), *Chloroflexi* (OTU_40961 and OTU_75010), and *Proteobacteria* (OTU_59288 and OTU_10968) had a high node degree (20> node degree >15) and played an important role (Appendix A). In the YQ network, all of the nodes had a smaller node degree, whereas OTU_29287, OTU_61084, and OTU_3172 had the top node degrees, with values of 7, 6, and 7, respectively (Appendix A). Moreover, all three nodes belonged to *Actinobacteria*. In the DT network, OTU_26953 (*Acidobacteria*), OTU_47804 and OTU_21400 (*Actinobacteria*), and OTU_3503 and OTU_33441 (*Chloroflexi*) had high node degrees, with values of 13, 13, 14, 11, and 11 (Appendix A). Compared to the node sizes of the other networks, the degree values of the YQ network were smaller (Figure 4). Furthermore, the dominant bacterial species for all four networks showed significant changes. These results implied that different mining areas were selected for different bacterial communities, which suggested that the interactions among different microbial taxa in the soil bacterial communities were substantially changed according to where they were located. This result confirmed the fact that the microbial distribution patterns across spatial distance and the interactions of the bacterial communities varied among the mining areas, which supported the third hypothesis.

The connectivity within and among modules has been reported in order to identify the roles of nodes in the MENs [29]. We used peripherals, connectors, module hubs, or network hubs to assign every node in the ecological networks. In the four networks, peripherals occupied >96% of the total nodes. Compared to one module hub (OTU_29287) in the YQ network, more module hubs appeared in the PB (OTU_2777 and OTU_13398), ZC (OTU_10968, OTU_8126, and OTU_59288), and DT (OTU_26953, OTU_21400, and OTU_40485) networks (Appendix A). We observed some connectors in the PB, ZC, and DT networks, while the YQ network did not have any connectors (Figure 5). Compared to the module hubs, we detected more connectors, especially in the PB network, which had nine connectors. Figure 5 also showed that the module hubs and connectors had a wide distribution in various microbial populations. Of the total nine module hubs, three belonged to *Acidobacteria*, three to *Actinobacteria*, one to *Chloroflexi*, and two to *Proteobacteria*. Nine connectors in the PB network belonged to the bacterial phyla *Acidobacteria*, *Actinobacteria*, *Bacteroidetes*, *Chloroflexi*, and *Proteobacteria*. Moreover, two connectors, which were *Actinobacteria* and *Chloroflexi*, were shown in the ZC network, whereas the two connectors identified in the DT network both belonged to phylum *Acidobacteria*. A notable phenomenon was that we did not identify a network hub in the four networks. The result suggested that *Acidobacteria* occupied the first percentage of the module hubs and connectors, then followed then by *Actinobacteria*, *Chloroflexi*, and *Proteobacteria*. Furthermore, phyla *Bacteroidetes* appeared just once.

As shown in Figure 6a, in the PB network, the nodes with a high degree belonged to modules 1 and 2, including *Acidobacteria* (OTU_24020, OTU_19752, OTU_20695, OTU_13398, and OTU_23653) and *Gemmatimonadetes* (OTU_30606 and OTU_23181). Notably, OTU_13398 also worked as a module hub. Moreover, OTU_19164, which was identified as the phylum *Nitrospirae* from module 2, had a high degree, in addition to OTU_27917 (*Proteobacteria*) and OTU_22409 (*Actinobacteria*). In the ZC network, the nodes with a high degree were primarily distributed in modules 1 and 2, which were identified as phyla *Chloroflexi* (OTU_40961 and OTU_75010), *Acidobacteria* (OTU_8126 and OTU_34138), and *Proteobacteria* (OTU_10968 and OTU_59288). OTU_8126 had the highest degree and worked as the module hub, although OTU_10968 and OTU_59288 also served as module hubs (Figure 6b). In the YQ network (Figure 6c), the nodes all had a small degree compared to the other three networks. OTU_29287, shown as *Actinobacteria*, had the highest degree and worked as the module hub. In the DT network, the nodes with high degree were primarily distributed in modules 1, 3, and 11, which were shown as phyla *Actinobacteria* (OTU_47804, OTU_21400, and OTU_40485), *Acidobacteria* (OTU_26953), and *Chloroflexi* (OTU_3503 and OTU_33441). Notably, OTU_21400 had the highest degree and worked as the module hub, whereas OTU_26953 and OTU_40485 played the roles of module hubs (Figure 6d).

### 3.4. Eigengene Network Analysis

Module 6 in Appendix A illustrated a conceptual example of eigengene network analysis (Appendix A). The eigengene network analysis was composed of various components. In module 6, for example, the heatmap showed the standardized relative abundances (SRAs) of bacterial species across 14 samples within module 6 in the PB network. In the heatmap, each row corresponded to the individual OTUs in module 6, whereas columns indicated the 14 samples in the PB network. The SRA of the corresponding eigengene (y-axis) across the samples (x-axis) were also shown in module 6. Appendix A showed that only five microbes had significant module memberships, where the y-axis shows the SRAs and the x-axis shows the individual samples. The values in parentheses are module memberships, and the module memberships included in the analysis correspond to the key species within a module. We examined module membership, which is shown as the square of the Pearson correlation between the given species abundance profile and the module eigengene. We identified significant module memberships within the respective modules (Appendix A).

In this study, there were 6, 10, 9, and 13 modules in the eigengene analysis of the PB, ZC, YQ, and DT networks, respectively. The module eigengenes explained 53%–81%, 61%–77%, 53%–81%, and 52%–70% of the variations in relative species abundance across the different samples in the PB, ZC, YQ, and DT networks, respectively (Appendix A). All of the eigengenes explicated over 50% of the observed variations, which revealed that these eigengenes could represent species shift across different samples in the individual modules.

The meta-modules were shown as groups of eigengenes in dendrogram in the eigengene network, which implied the higher-order structure of the constructed network. In this study, the eigengenes from the modules showed significant correlations. Many meta-modules were clustered for the ZC, YQ, and DT networks, whereas only one meta-module was clustered for the PB network (Figure 7a). The eigengenes from the paired modules were clustered differently in the different networks, which implied that the higher order organization of the paired modules was totally different among the different mining areas. Otherwise, to check which property was most important for the network modules, we investigated the trait-based module significances, by squaring the correlation between signal intensity of the modules and some soil characteristics, including the climate parameters of AAP (annual average precipitation) and AAT (annual average temperature; Figure 7b). Figure 7b showed that strongly significant or significant correlations existed for the PB network between the connectivity of the five modules and the selected variables, including AAT, pH, SOM (soil organic matter), and AN (ammonium nitrate; *p* ≤0.001, 0.001 ≤ *p* ≤ 0.05). For the ZC network, the connectivity of only one module was significantly related to the AAT and EC (electrical conductivity) values (0.001 ≤ *p* ≤ 0.05). No significant correlations were observed, however, between the connectivity of all the modules and the properties in the YQ network (*p* > 0.05). For the DT network, the connectivity of three modules showed significant correlation with the selected variables, such as pH, AP (available phosphorus), and AK (available potassium; 0.001 ≤ *p* ≤ 0.05). Moreover, these results indicated that these properties, which correlated with the keystone bacterial community were totally different in different mining areas, thus supporting the second hypothesis.

On the other side, a Mantel test and correlation test were performed to screen for the dominant environmental factors, which affected the soil microbial community structure. The results were shown in Appendix A. The Mantel test showed that microorganisms were closely correlated with AAP, AAT, EC, NN, AP, and AK (*p* < 0.05; Appendix A). According to the Pearson correlation coefficient and significance, a correlation test presented the results that, AAP, AAT, EC, AN, AP, and AK had significant impacts on the structural differentiation of bacterial compositions (Appendix A). 

According to above results, CCA and VPA were performed to analyze the correspondence between the environmental factors and microbial community groups for the four mining areas (Figure 8). As shown in Figure 8a, samples from the ZC and YQ groups were almost gathered, while PB and DT were separated with them. The correlation information between environmental factors and communities can be expressed by the angle between the environmental factor arrow line and the linking line, which connected the sample points and center points. Therefore, on the left part of Figure 8a, the correlations between AAT and PB communities were the largest compared to the EC value and AK, while AP had a closer relationship with YQ and ZC groups. Based on the results of CCA, AAT presented the highest explanation percentage for the analysis between environmental factors and microbial communities, followed by AP > AK > EC (Figure 8b). 

## 4. Discussion

The rapid development of technologies, such as high-throughput sequencing technologies, has provided a huge amount of scientific data, especially in the field of molecular ecology [30]. Moreover, dealing with a huge amount of data, as well as using these data to understand functional processes at the community level, presents significant challenges. Moreover, the network interactions play an important role in the ecosystem processes and functions. Therefore, based on high-throughput sequencing data, we constructed several networks, investigated the different interactions in the microbial communities of the semihumid and semiarid mining ecosystems, and identified the key populations. We also examined the relationships between network structures and soil properties.

The results of the taxonomic composition of microbial consortia in different mining areas (Table 1, Figure 2) suggested that, the observed species changed across long distances, which might signify that some new species were generated with different locations. While the disturbed mining soil environment might pose a challenge for some species in the soil, it still stimulated new microbial species, especially for bacteria that can adapt to special, reclaimed environments. From the semihumid locations to the semiarid areas, soil microbial diversity showed the decreasing trends, which might suggest that some special environment might be formed in the damaged, regional mining sites, and could cause some bacteria to die. Most soil microbial phyla represented in this study belonged to 13 major phyla. Nevertheless, the species distributions on the phylum level were different for the four mining areas. Regarding the temporal variation in Figure 3, the PCA and NMDS results indicated that the microbial communities changed throughout the spatial distance. Figure 3 displayed that the PB group was far away from the DT group, while the distance was slight between ZC and YQ. While the percentage of the PC2 (principal component) explanation was just 11%, the result obtained on this axis might still be reliable in interpretation (Figure 3a). This result, in Figure 3b, might indicate that the microbial structure has become more different with locations changing from semihumid to semiarid mining areas, which was in line with the research investigated by Helingerová et al. research [17]. Moreover, the response ratio calculation (RRC), and linear discriminant analysis effect size (LEfSe) methods were also used to analyze the differences among the soil community structures in the four mining areas (Appendix A). The RRC and LEfSe results also implied the existence of gaps in the four groups, which confirmed that the observed changes of the microbial community were significantly impacted by the changing spatial locations. Moreover, in spite of the soil microbe being significantly disturbed or destroyed by mining activities, especially in the semiarid areas, the soil microorganism, through interactions, might be resuscitated or restored on their own.

In this study, we also analyzed the microbial interactions in different mining areas using the method of molecular ecological network analysis. The network properties changed across all four mining areas (Table 2), from semihumid to semiarid areas, and the species involved in the microbial interactions also changed, as demonstrated by variations in the dominant phyla (high node degree) (Figure 4 and Figure 5). In the networks, community stability was higher with increased complexity. The simple network structure (no connectors or module hubs and more sparsely distributed species) and low competitive connections may have caused a negative effect on biogeochemical functions, indicating an unstable and vulnerable microbial community when other disturbances occurred. Microorganisms under this condition might have been specialized to the local environments and may thus have been sensitive to environmental changes. The results (Table 2 and Figure 4) indicated that the network interactions for some microbial groups were more complicated in the PB network located in the semihumid area, although the microbial community diversity in this network was the poorest. The nature parameters in PB and ZC are totally different from those of YQ and DT. This result implied that different natural condition might have significantly affected the microbial community structure and their network interactions in different ways.

The networks obtained showed the general features of many cellular networks, such as modular, small world, or scale-free [31]. A small-world pattern contributed to the efficient communication of different members in a community, and could quickly respond to external environmental changes, such as mine subsidence, subsidence cracks, landslides, or soil reclamation. Closeness centrality is based on the average shortest paths, and thus reflects the central importance of a node in disseminating information. Complex networks with greater connectivity are more robust to environmental perturbations than simple networks with lower connectivity [32,33]. In this sense, the higher complexity of the PB and ZC networks suggested that (Table 2), as different taxa were complementary, the microbiome in the eastern semihumid mining areas with a high groundwater level was more resilient to environmental stresses, such as mine subsiding or land reclamation activities. The result might imply that the network structural complexity might be related to the geographic location and environment. Further studies are necessary to corroborate this observation.

We considered the OTUs with the highest degree and highest closeness centrality, and the lowest betweenness centrality scores to be the keystone taxa [34]. Keystone taxa are highly connected taxa that play important roles in the microbiome, and their removal can cause significant changes in microbial composition and functioning [35]. Although previous studies have reported keystone taxa in various environments, reports on keystone taxa in the disturbed mining areas have been limited [36,37,38,39]. As found in this study, key populations can be distinguished according to their network profiles and module memberships. Networks in the semihumid mining areas, with a high groundwater level, such as PB and ZC, showed that the keystone taxa belonged to the microbial phyla *Acidobacteria, Gemmatimonadetes, Chloroflexi*, and *Proteobacteria*, whereas *Actinobacteria, Acidobacteria*, and *Chloroflexi* were the key species in the YQ and DT networks from semiarid mining areas (Figure 3, Figure 4 and Figure 5; Appendix A). Although the YQ mining area is far away from the PB and ZC mining areas, the most abundant phylum was the same (i.e., *Proteobacteria*), whereas *Actinobacteria* was the most abundant in the DT mining area, despite the fact that the two mining areas were closer to each other (Figure 1). This result might indicate that no direct relationship existed between the location sites and microbial abundance. We know that *Proteobacteria* is widely distributed around the world. It has an aerobic bacterium that is capable of degrading a variety of contaminants, as well as some bacteria that produces several oxidases that oxidize diverse compounds [40]. *Proteobacteria* has a highly diverse physiology and is distributed in almost all of the different ecological environments. Mining areas are complicated and contain surface subsidence, cracks, landslides, reclamation, and restoration areas. This complicated condition might result in a suitable environment for *Proteobacteria*, which may have made *Proteobacteria* the dominant bacteria. *Actinobacteria* are ubiquitous gram-positive bacteria, and have a characteristic filamentous morphology, which might be the reason for their high abundance in the semiarid DT mining area. In addition, *Actinobacteria* have a variety of important functions that make them useful and powerful in soil and marine environments, including degradation of organic substances. In the adverse and comprehensive semiarid mining areas, the existence of *Actinobacteria*, as the dominant microbe, might help to improve the soil quality. *Acidobacteria* play a significant role in soil ecological processes, and this diverse phylum is widely distributed throughout various natural environments [41,42].

On the other side, the abundances of key taxa *Chloroflexi* and *Gemmatimonadetes* were low, suggesting the lack of a direct relation between abundance and key functional importance. Chen et al. (2017) have reported that *Chloroflexi* increased when the environment became more anaerobic. It is possible that mining areas have many kinds of environments, such as surface subsidence or cracked areas, which are suitable for the *Chloroflexi* [43]. Even though the ecological function of *Chloroflexi* was not clear, this phylum was still the keystone microorganism in the four mining areas. Recently, phylum *Gemmatimonadetes* has been described as a bacterial group whose members are widespread in soil habitats. Its cultured representative genus is *Gemmatimonas aurantiaca*, which has been isolated and is able to grow under not only anaerobic conditions but also aerobic conditions [44,45]. This finding might suggest that *Gemmatimonadetes* could be a suitable phylum in complicated mining areas that contain aerobic and anaerobic environmental habitats. This might be the reason why the *Gemmatimonadetes* was the keystone taxa. Tobin-Janzen et al. [46] reported that *Nitrospira* was the dominant genus of bacteria in soil samples from an underground coal-mining fire (Pennsylvania, USA). Sun et al. [47] have found a similar conclusion, that is, *Nitrospira* accounted for the highest proportion in the soil samples from China. However, what these past studies found was different from the results from our study. Ezeokoli et al. [48] have investigated the microbial community in opencast coalmines but did not study the keystone taxa. Their results showed that microbial communities in mining areas have been impaired and have had negative effects on soil biological processes, especially nutrient cycling and ecosystem sustainability.

In this study, the connection between two OTUs indicated that the two OTUs might respond to a common environmental parameter. Then characterization of the OTU connections in modules could be used to describe these interactions among the microbial communities [49]. Additionally, it might be suggested that the same underlying factors motivated changes in OTU abundances with strong module memberships. Therefore, OTUs with strong module memberships should have some physical or functional relationships in the community. As shown in this study, module memberships, topological roles, and phylogenetic relationships have provided some information to identify the key OTUs. Thus, the interactions and ecological roles of these microbial communities in mining areas might provide insight for mining activities in China, especially for ecologically fragile and vulnerable areas. For the first time, this study presented different network interactions among soil microbial communities in semihumid mining areas with high groundwater levels and semiarid mines.

In fragile ecological systems, understanding how the soil microbial communities respond to external environmental changes, in particular, for anthropogenic change, is significantly important [50]. In this study, the method of network analysis revealed an appropriate way to discover how environmental changes affected microbial communities. Previous studies have shown that when the external environment changed, such as variations in soil properties, the diversity of microbial communities changed, which may be correlated with disturbances in soil characteristics [51,52,53]. Additionally, soil factors, such as pH, moisture content, total carbon content, and organic matter, have been reported to have a greater impact on the soil bacterial community structure and diversity in the ecological restoration of mining areas. For example, Xiao et al. [54] have reported that soil microbial activity was affected by soil factors to different degrees, and that soil microbes played a critical role in the recycling of soil nutrients and soil fertility. Pille Da Silva et al. [55] found that soil microbiological attributes affected microbial biomass carbon and microbial basal respiration. The microorganism could increase soil quality and restore biological diversity in the coal-mining area. Understanding this relationship between a microbial community and soil properties is critical to the ecological restoration of coal-mining areas. Bi et al. [56,57] found that the arbuscular mycorrhizal (AM) fungal community was influenced by the mine slope position and subsidence. Their study clarified that the AM fungal ecological function could potentially aid vegetation restoration and reduced erosion in coal-mining areas.

In our study, we identified strongly significant or significant correlations between the node connectivity in the module and the selected environmental variables, such as AAT, soil pH, SOM, and ammonium nitrate content, for the PB network. Fernández-Montiel et al. [58] showed that soil pH could change the slightly acidic environments to an acidic condition. Results from 12 sites, following the mining activities of different lengths of time, in terms of reclamation, suggested that soil microbial abundance, taxonomic diversity, and functional diversity could be improved by increasing the number of reclamation years [59]. A redundancy analysis revealed that soil pH was significantly important in microbial metabolic structure and bacterial genetic assemblages. This finding was similar to our results, that is, the soil pH value was significantly correlated to different species of module 6, but only in the PB network (Figure 7), especially for the OTU_19170 (*Koribacteraceae*), which belonged to *Acidobacteria* (Appendix A). Sáenz de Miera et al. [60] presented the finding that subgroups of *Acidobacteria* showed a significantly positive relationship with soil pH value. Soil organic matter is always represented as an important indicator to estimate the soil carbon storage and to evaluate soil quality. Disturbances introduced by mining activities might affect the activity of soil microbes, thus affecting the SOM content. The results in this study also showed that the SOM had a significant relationship with module 6, of which the important nodes OTU_4611 (*Burkholderiales*) and OTU_8175 (*Burkholderiales*) belonged to phylum *Proteobacteria*, which was the keystone phylum in the PB network (Appendix A). Therefore, all of these results may have suggested that the pH value and SOM revealed a complicated relationship with soil microbial communities, in particular with the keystone species in the PB network. In a sense, these results confirmed that the method of network analysis was effective and feasible to analyze the relationship between environmental factors and microbial community structures.

The AAT and EC value were significantly related to two modules in the ZC network. During the succession of land following coal mining, aggregate stability and organic matter increased, whereas the EC value decreased. Other researchers have examined the soil bacterial characteristics of 21 coal-mining sites [61]. One result was that the bacterial species composition was significantly correlated with the soil EC value, which was similar to conditions in the ZC network. Our results showed that the soil EC value was significantly correlated with different species of module 1 (Figure 7), especially for the OTU_34138 and OTU_8126 (*Acidobacteria*), which belonged to keystone species in the ZC network (Appendix A). The EC value presented as a kind of soil-leaching solution, which reflected the water-soluble salt content in the soil. Once the soil was disturbed by mining activities, the solubility of calcium carbonate or magnesium carbonate in the soil might have been affected, and then the water-soluble salt content in the soil-leaching solution changed, which influenced the microbial communities. Sun et al. [47] have found that the distribution of bacteria was primarily affected by SOM, AK, and AP in similar coal-mining areas. The location map in Figure 1 shows that PB and ZC are close to each other, and belong to the semihumid area. Moreover, both of them are located in the coal-mining areas with high groundwater levels. For these areas, the soil was affected by a secondary anti-alkali and heavy metal migration problems. In the future, we need to investigate additional properties, such as heavy metal contents.

In the DT network, the soil variable pH and AP showed a significant correlation with the module, whereas we did not identify a significant correlation between the modules and soil variables in the YQ network. This result implied that pH and AP values might have played an important role in the DT network structure. In the DT network, pH showed a positive relationship with the phyla *Acidobacteria*. Notably, the important nodes in module 7 all belonged to *Acidobacteria* (Appendix A). This suggested that *Acidobacteria* was significantly correlated with the soil pH value. Ma et al. [40] have reported similar results, that is, the abundance of *Acidobacteria* changed with variations in the soil pH value. In this study, this result indicated that external environmental variables affected the network interactions among different microbial groups and that such changes may be related to soil properties, such as the pH value. These results also indicated that both pH value and mine activities affected the microbial and network structures.

Furthermore, AP showed a significant correlation with module 12 in the DT network, and the important nodes in module 12 (Figure 7) belonged to *Gemmatimonadetes* (OTU_34734, OTU_13954, and OTU_19203) and *Chloroflexi* (OTU_129; Appendix A). This indicated that *Gemmatimonadetes* and *Chloroflexi* were significantly correlated with the soil AP value and that *Chloroflexi* was the keystone species in the DT network. Furthermore, the presence of microbes in the same module implied that these microbial populations compartmentalized with each other to survive in response to disturbances caused by mining activities. It is well known that phosphorus is one of the most indispensable nutrient elements for soil development. It can be easily fixed in the soil, although its utilization rate is low. Moreover, phosphorus is a necessary element for microbial metabolism—for example, some soil microorganisms may produce acidic substances through metabolism, and then dissolve some insoluble phosphates and apply them to their own metabolic processes. All of these results could indicate that the soil phosphorus content might be correlated with keystone species. We speculated that the disturbed environmental factors influenced the microbial composition, thus influencing the AP content. In this study, we did not find any significant correlation between environmental factors and network modules in the YQ network, implying that we need to examine and include additional environmental factors in this analysis, or developed a new method to prove this relationship.

The combined results of the mantel test, correlation test, CCA, and RDA show that, in spite of the fact that environmental variables such as AP, AK, and EC showed significant effects on the microbial communities, the explanation percentages in the VPA plot (Figure 8) were very low. However, the natural factor, AAT, could explain 13.725% of the effect, which might suggest that natural geographic conditions influence the microbial community structures. On the other side, soil pH value and SOM are well known as being the key environmental factors that affect the soil bacterial communities [51,58,59,60]. However, on the phylum level, pH and SOM showed no effects on the microbial community structures, which might imply that natural geographic factors, such as the spatial distance (from semihumid to semiarid locations), play key roles in soil microbial compositions. 

Information on the common presence of bacteria related to keystone microbes, however, is still insufficient for these networks. We are still not able to identify the exact keystone species and their differentiations between the semihumid and semiarid mining areas, which may have mitigated the effects of soil disturbance and accelerated the restoration of mined soil. Moreover, the high-throughput 16S RNA gene sequencing only provided extensive information about only the taxa present in bacterial communities in disturbed mining areas, but did not provide enough insights into the functional roles of these keystones, which is essential for ecological restoration. Using Geochip technology, the ecological function network analysis and more extensive research on metabolism should be investigated in the near future.

## 5. Conclusions

This study demonstrated microbial interactions and their relationships in semihumid and semiarid disturbed mining areas. The results showed that soil bacterial compositions and the network interactions were completely different across the semihumid and semiarid mining areas. The results of keystone species suggested that different mining areas selected different microbial communities in order to resist the adverse environment. The results of trait-based module significances showed that several environmental factors (e.g., AAT, pH, EC, AP, and AK) were significantly correlated with some keystone OTUs. This study provided a new method to study network interactions among different microbial populations in different fragile ecological systems. Our findings also provided insight into the ways in which microorganisms responded to mining activities and changed their resilience by regulating their interactions in the significantly different ecosystems. In the future, more studies will be conducted on the functional network analysis to deepen our understanding of these mechanisms.

## Figures and Tables

**Figure 1 microorganisms-08-00433-f001:**
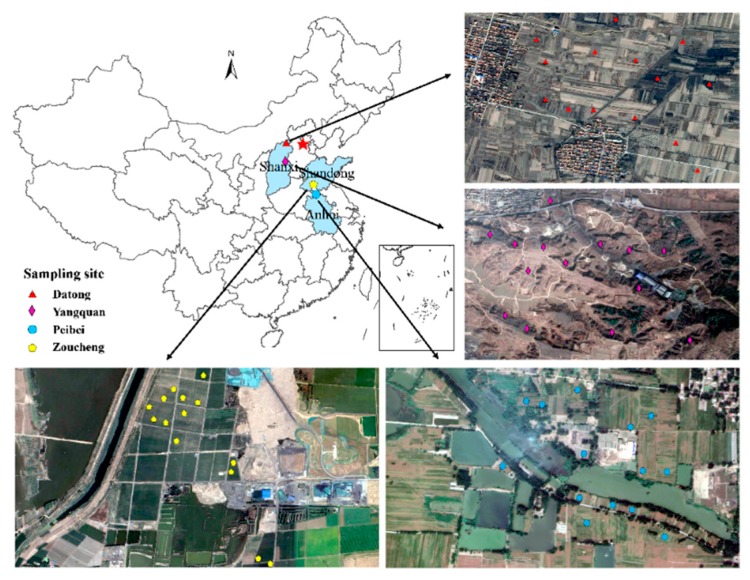
Location of the study area.

**Figure 2 microorganisms-08-00433-f002:**
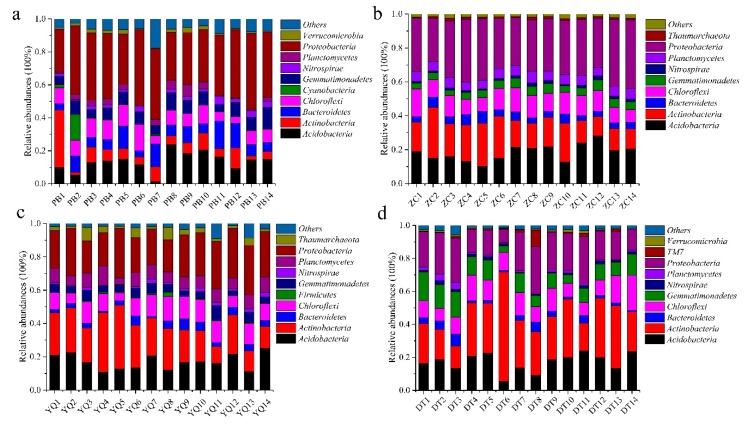
Bacterial categories (**a**–**d**) in soil samples across Peibei (PB), Zoucheng (ZC), Yangquan (YQ), and Datong (DT) mining areas.

**Figure 3 microorganisms-08-00433-f003:**
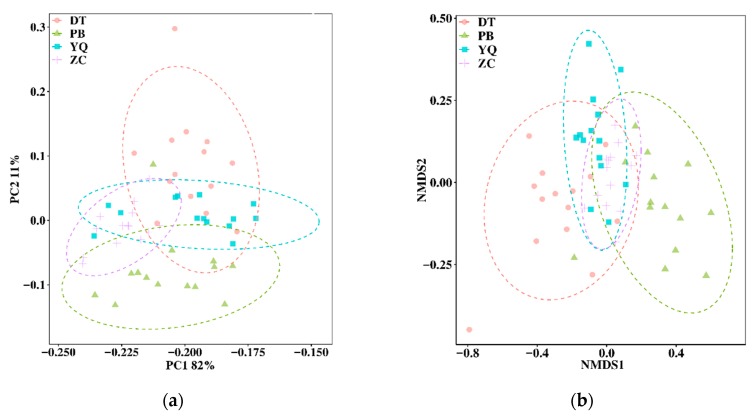
Principal component analysis (PCA; (**a**)) and non-metric multidimensional scaling (NMDS; (**b**)) analysis results of soil bacteria at the phylum level in mining areas.

**Figure 4 microorganisms-08-00433-f004:**
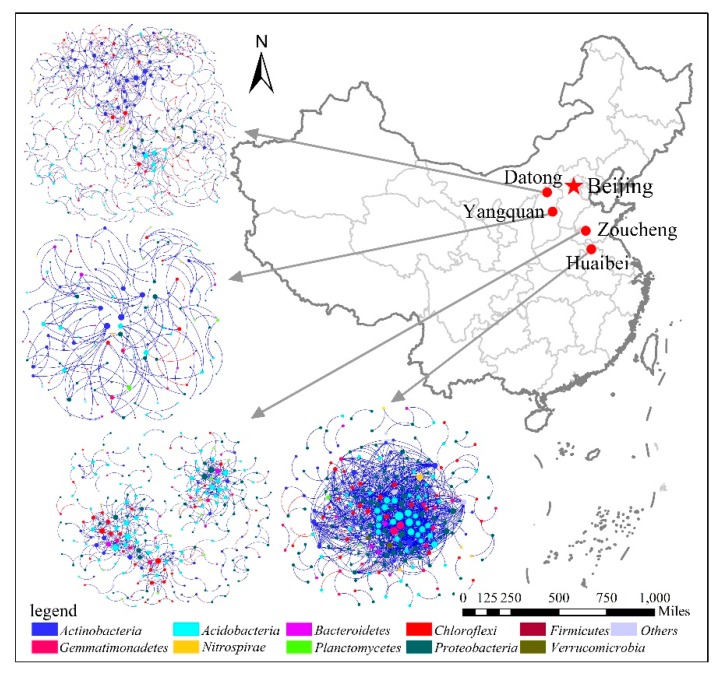
Overview of the networks in different mining areas, with node sizes being proportional to node degrees. A red link means a negative correlation and a blue link means a positive correlation.

**Figure 5 microorganisms-08-00433-f005:**
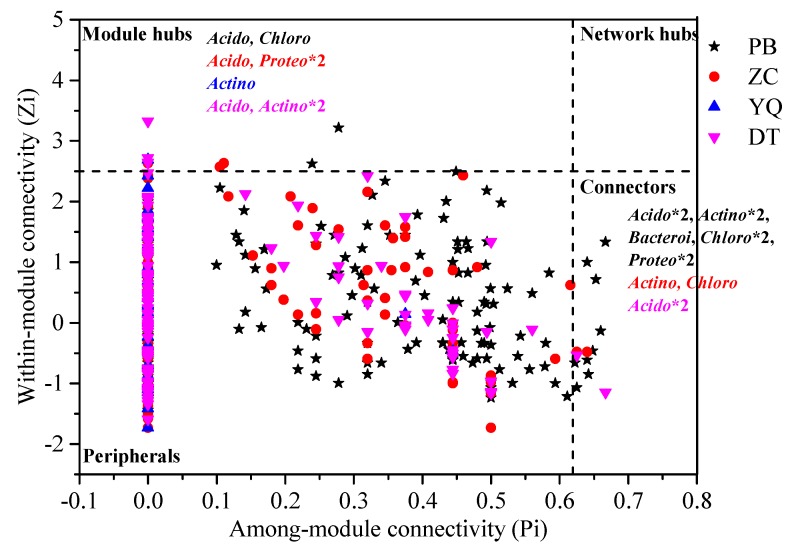
Z–P plot showing the keystone species in the different mining area networks. Different symbols with special colors represent different networks as follows: black star for the PB network, red circle for the ZC network, blue upward-facing triangles for the YQ network, and rose downward-facing triangles for the DT network. The module hubs and connectors are labeled with phylogenetic affiliations (*Acido—Acidobacteria, Actino—Actinobacteria, Bacteroi—Bacteroidetes, Chloro—Chloroflexi*, and *Proteo—Proteobacteria*. *2 means that there are 2 connectors or 2 module hubs belong to that phylum).

**Figure 6 microorganisms-08-00433-f006:**
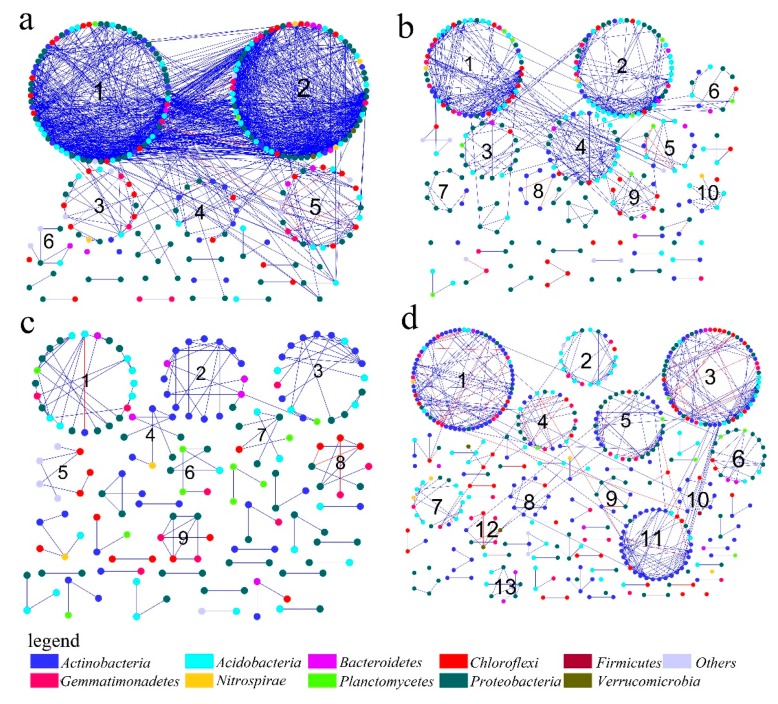
Network graph with module structure produced by the fast-greedy modularity optimization method. Each node corresponds to a microbial population. Arable numbers such as 1, 2, 3, and 4 stands for the module number. A red link indicates a negative correlation, and a blue link indicates a positive correlation. (**a**–**d**) stand for the PB, ZC, YQ, and DT network, respectively.

**Figure 7 microorganisms-08-00433-f007:**
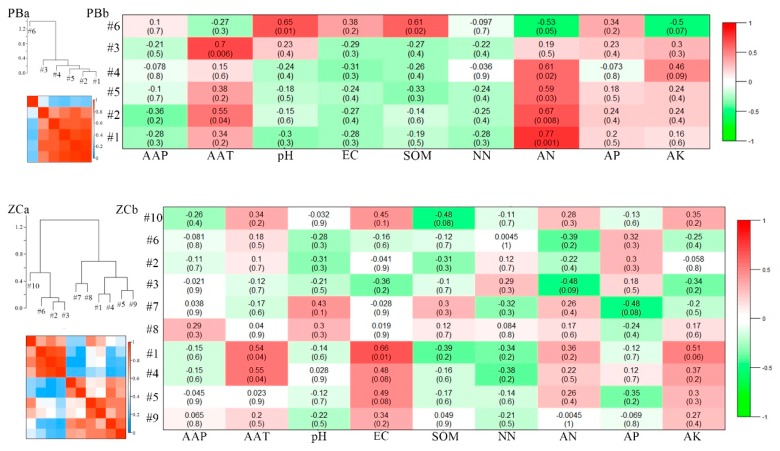
Correlations and heatmap of module eigengenes of the four networks (**a**). Correlations between the signal intensity of a module and each soil characteristic for the four networks (**b**).

**Figure 8 microorganisms-08-00433-f008:**
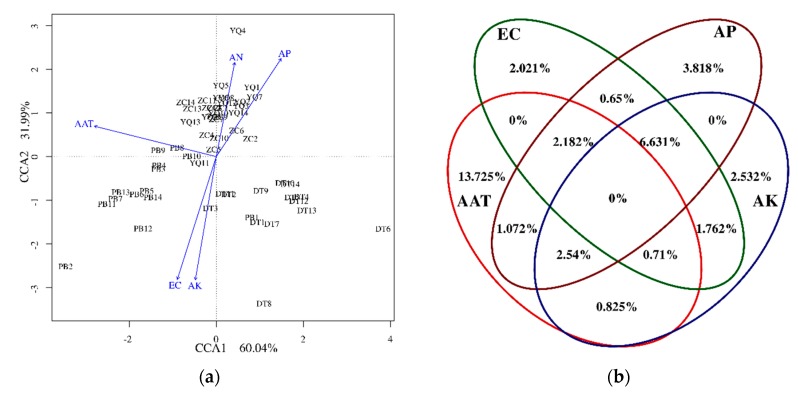
Correspondence analysis (CCA; (**a**)) and variation partition analysis (VPA; (**b**)) plots indicate the relationships between the bacterial community and soil properties.

**Table 1 microorganisms-08-00433-t001:** The alpha diversity index of soil microorganisms in the four mining areas.

Mining Area	Chao	Shannon	Pielou Evenness
PB	3710.10 ± 836.20 ^b^	6.5937 ± 0.5738 ^ab^	0.8305 ± 0.0441 ^c^
ZC	8319.38 ± 541.91 ^ab^	7.7519 ± 0.0824 ^c^	0.9014 ± 0.0105 ^bc^
YQ	4917.14 ± 891.56 ^c^	6.8408 ± 0.2617 ^bc^	0.8463 ± 0.0177 ^ab^
DT	5238.99 ± 421.43 ^bc^	6.7701 ± 0.1585 ^b^	0.8120 ± 0.0144 ^ab^

Mean ± standard deviation (SD) (*n* = 14). Superscript letters such as a, b and c, indicate significant differences between different sampling sites for each parameter separately using Duncan test at significant *p* < 0.05 level (ANOVA analysis, *n* = 14). PB, ZC, YQ and DT stand for the sampling sites Peibei, Zoucheng, Yangquan and Datong.

**Table 2 microorganisms-08-00433-t002:** Topological properties of the empirical molecular ecological networks of microbial communities and their random networks in different mining areas.

	Network Indexes	PB	ZC	YQ	DT
Empirical networks	Similarity threshold	0.86	0.86	0.86	0.86
	*R*^2^ of power law	0.837	0.931	0.852	0.896
	Total nodes	248	265	165	441
	Total links	1285	516	163	640
	Average degree (avgK)	10.363	3.894	1.976	2.902
	Average clustering coefficient (avgCC)	0.314	0.258	0.158	0.184
	Average path distance (GD)	3.334	7.725	3.975	7.802
	Modularity	0.364	0.701	0.897	0.829
	Module number (with >5 nodes)	6	10	9	13
Random networks	Average clustering coefficient (avgCC)	0.134 ± 0.010	0.028 ± 0.006	0.007 ± 0.005	0.008 ± 0.003
	Average path distance (GD)	2.772 ± 0.024	3.877 ± 0.058	6.454 ± 0.448	5.022 ± 0.076
	Modularity	0.228 ± 0.005	0.496 ± 0.008	0.795 ± 0.011	0.637 ± 0.008

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
