# Peer review of "Molecular Ecological Network Complexity Drives Stand Resilience of Soil Bacteria to Mining Disturbances among Typical Damaged Ecosystems in China"

_microorganisms, 2020, doi:10.3390/microorganisms8030433_

Round 1

Reviewer 1 Report

This manuscript reports a study on the bacterial community composition in different mining areas in China.

English language needs to be revised throughout the manuscript. I recommend that the manuscript is checked by an English native speaker.

L22: “Bacterial community diversity and composition classified measurably between different sites.”. Something is missing from this sentence.

L24: Bacterial phyla (e.g. Proteobacteria, Acidobacteria, etc.) should be italicised. Please check throughout the manuscript text, figures and tables.

L26: “some environmental factors”. Please be specific.

L27: “The keystone species in different mining areas suggested that selected different microbial communities to resist the adverse environment.”. This sentence is unclear.

L30: “correlated with some keystone species”. Please be specific.

L37: “Globally, coal mining has resulted in surface subsidence and has created the ecological environment more fragile by creating huge overburden dumps and voids”. This is an example of a poor English sentence.

L65: What do the authors mean by “massive biological species”?

L68: “And the ability to elaborate and analyze the interactive network structures and the underlying mechanisms is essential to study microbial biodiversity.” Another example of a poor English sentence.

L88: “Some studies also reported how soil bacterial community structure and diversity changed after coal-mining disturbances [18].” Authors refer to “some studies”, yet only one reference is provided.

L94: replace “this data” with “these data”.

L117: microflora is an outdated term. Please consider using microbiome.

L663: “several environmental factors”. Which factors? Please be specific?

Author Response

Comments and Suggestions for Authors

This manuscript reports a study on the bacterial community composition in different mining areas in China.

Response: We gratefully thank the reviewer for revising our manuscript. Comments and suggestions have been mostly taken into account in the revised version of the manuscript. We also response to each specific comment.

English language needs to be revised throughout the manuscript. I recommend that the manuscript is checked by an English native speaker.

Response: We thank the reviewer for the suggestion. We have invited an English native speaker to help with this paper, and polished this paper on the web of MDPI.

L22: “Bacterial community diversity and composition classified measurably between different sites.”. Something is missing from this sentence.

Response: Thanks for the suggestion. Taking into account this observation, we modified the sentence into “Bacterial community diversity and composition were classified measurably between semi-humid and semi-arid damaged mining sites.” (L. 22-23)

L24: Bacterial phyla (e.g. Proteobacteria, Acidobacteria, etc.) should be italicised. Please check throughout the manuscript text, figures and tables.

Response: Thank you very much for the valuable suggestion. We have checked throughout the whole paper to make sure the correct expression, especially in the legends of figures shown as below.

L26: “some environmental factors”. Please be specific.

Response: Thank reviewer for the suggestion. Taking into account this observation, we added the environmental factors, such as annual average temperature, pH value, electrical conductivity value, and available phosphorus value. (L. 26-27)

L27: “The keystone species in different mining areas suggested that selected different microbial communities to resist the adverse environment.”. This sentence is unclear.

Response: Thanks for the valuable suggestion. Taking into account, we modified the sentence into “The keystone species in different mining areas suggested that selected microbial communities, through natural successional processes, were able to resist the corresponding environment.” (L. 28-30)

L30: “correlated with some keystone species”. Please be specific.

Response: Thank reviewer for the suggestion. Taking into account this observation, we have presented some keystone species, such as OTU_8126 (Acidobacteria), OTU_8175 (Burkholderiales) and OTU_129 (Chloroflexi). (L. 32-33)

L37: “Globally, coal mining has resulted in surface subsidence and has created the ecological environment more fragile by creating huge overburden dumps and voids”. This is an example of a poor English sentence.

Response: We thank the reviewer for the valuable suggestion. We have polished the English language of whole paper on the web of MDPI. (L. 40-41)

L65: What do the authors mean by “massive biological species”?

Response: Thank reviewer for the suggestion. We have modified the expression to “massive microbial species” means lots of microbial species in the soil microbial communities. (L. 68)

L68: “And the ability to elaborate and analyze the interactive network structures and the underlying mechanisms is essential to study microbial biodiversity.” Another example of a poor English sentence.

Response: We thank the reviewer for the valuable suggestion. We have polished the expressions throughout whole paper on the web of MDPI. (L. 71-73)

L88: “Some studies also reported how soil bacterial community structure and diversity changed after coal-mining disturbances [18].” Authors refer to “some studies”, yet only one reference is provided.

Response: Thank reviewer for the suggestion. We have provided more references. (L. 92)

L94: replace “this data” with “these data”.

Response: Thanks for the suggestion. We have modified the expression to “these data”. (L. 97-98)

L117: microflora is an outdated term. Please consider using microbiome.

Response: Thanks for the suggestion. We have modified the expression to “microbiome”. (L. 120)

L663: “several environmental factors”. Which factors? Please be specific?

Response: Thank reviewer for the suggestion. Taking into account this observation, we have presented environmental factors, such as AAT, pH, EC, AP and AK. (L. 666)

Reviewer 2 Report

The authors tackle an interesting question, namely which factors govern the recovery of soil after mining. But the study lacks a proper study design by which the data and the interesting data evaluation would have become meaningful. This could have been either a time course on one site or more than 20 sites that differ in only a few parameters, like e.g. a gradient of aridity.

Author Response

Comments and Suggestions for Authors

The authors tackle an interesting question, namely which factors govern the recovery of soil after mining. But the study lacks a proper study design by which the data and the interesting data evaluation would have become meaningful. This could have been either a time course on one site or more than 20 sites that differ in only a few parameters, like e.g. a gradient of aridity.

Response: We gratefully thank the reviewer for reviewing our manuscript. Comments and suggestions have been mostly taken into account in the revised version of the manuscript.

In this study, we have selected just 4 mining areas. We have modified the expression in the part of materials and methods. The reasons for this comment are shown as below. As well know that there are many mines in the semi-humid and semi-arid areas of China, and many of them are closely to each other. For this reason, in this study, the selected 4 mines are only used as the representative objects of the geographical environment in the Eastern and northern-western China. We have also finished some investigations in many other semi-arid mining areas (e.g., Daliuta mine in Shanxi provine, Heidaigou mine in Inner Mongolia province) in 2019 year, and we will continue to conduct more mining areas in the near future, in order to accumulate amount of research works to improve this meaningful topic.

For the English language, we thank the reviewer for the valuable suggestion. We have invited an English native speaker to help to revise and polish this paper on the web of MDPI.

Round 2

Reviewer 1 Report

All the raised issues have been satisfactorily addressed.